# Optical Diffraction-based Convolution for Semiconductor Mask Optimization

## Abstract

In recent years, the increasing demand for smaller and more powerful semiconductors highlighted the critical role of lithography—a key stage in semiconductor manufacturing responsible for precise mask design and wafer patterning. To meet these demands, the semiconductor industry has increasingly adopted computational lithography, employing machine learning and deep learning techniques to accelerate advancements in lithographic technology. Despite the various research efforts and successes in computational lithography, there remains a lack of explicit incorporation of physical principles. This gap limits the ability of existing methods to fully capture the complex physical phenomena inherent in lithography behaviors. To bridge this gap, we propose **OptiCo**, a novel convolutional neural network that seamlessly integrates optical diffraction principles into its architecture. At its core, OptiCo employs an *optical phase kernel* to model phase variations resulting from light propagation, effectively capturing the physical interactions among light, masks, and wafers. We evaluate OptiCo on semiconductor lithography benchmarks, demonstrating its superior performance in mask optimization tasks, with its remarkable generalization capabilities in OOD datasets.

## 1 Introduction

Semiconductors are now indispensable to almost every facet of modern life, enabling technologies from smartphones and computers to artificial intelligence systems. Their critical role has brought semiconductor manufacturing to the forefront of technological innovation and industrial focus. Within this manufacturing ecosystem, lithography stands out as one of the most important stages (Moreau, 2012; Zheng et al., 2023a; Yang et al., 2022b). In particular, the lithography process uses light passing through a mask to precisely project the patterns of transistors and circuits onto silicon wafers. Due to its technical demands and the precision required, the lithography stage alone accounts for approximately 30% of the overall manufacturing cost (Ma & Arce, 2011).

Over the years, the increasing demand for smaller and more powerful semiconductors has pushed lithography technology to its physical limits. As manufacturers strive to further reduce circuit sizes, they face substantial challenges from the complex interactions between light, wafer, and mask patterns (Bakshi, 2009; Braam et al., 2019; Giannopoulos et al., 2024). As shown in Figure 1, the image projected from the mask onto the wafer typically deviates from the intended design due to the effects of light diffraction, which are particularly pronounced with short-wavelength light (e.g., EUV light) used for fabricating smaller circuit feature sizes (Wu & Kumar, 2007).

To achieve the precision in lithography, traditional approaches have relied on trial-and-error experimentation or mathematical modeling to optimize lithographic parameters (Mack, 2005; Gao et al., 2014; Yu et al., 2022; Yang & Ren, 2025). However, repeatedly executing the real lithography process is prohibitively expensive, and the complexity of light diffraction makes mathematical modeling both challenging (Ma & Arce, 2011; Banine et al., 2011). To address these constraints, the semiconductor industry has adopted simulation-based computational lithography, particularly those leveraging deep learning (DL) methods (Banerjee et al., 2013; Siemens; Watanabe et al., 2017).

Among the various DL-based approaches, GAN-OPC (Yang et al., 2018) emerged as one of the earliest DL frameworks that utilize a generative adversarial network (GAN) to establish a direct mapping between input target wafer patterns and output mask designs. Building upon this foundation, DAMO (Chen et al., 2020) enhanced performance by achieving higher-resolution mask predictions by

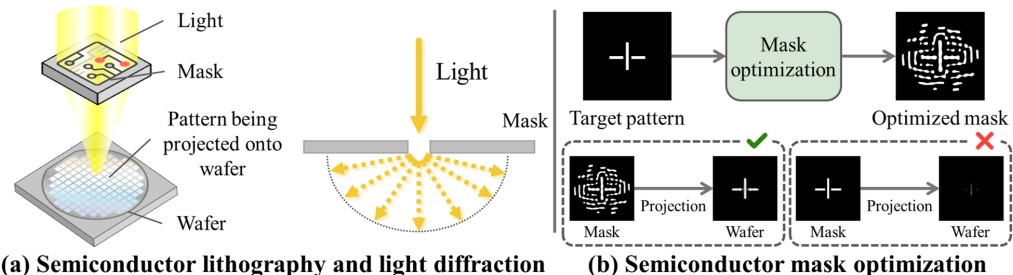

**(a) Semiconductor lithography and light diffraction**    **(b) Semiconductor mask optimization**

Figure 1: Illustrations of semiconductor lithography and mask optimization.

incorporating a UNet++ (Zhou et al., 2018) backbone and residual blocks. TEMPO (Ye et al., 2020) further leveraged multi-domain image-to-image translation techniques to address the complexities associated with 3D mask designs.

Despite these achievements, some studies (Yang et al., 2022a; Yang & Ren, 2023) have highlighted limitations in their reliance on standard neural network architectures, which may struggle to capture the global information essential for accurate lithography modeling (Hopkins, 1951). To overcome this limitation, researchers have explored the potential of Fourier Neural Operators (FNOs) (Li et al., 2020). Specifically, DOINN (Yang et al., 2022a) introduced an FNO-based framework that employs Fourier transforms to effectively capture both low-frequency global information and high-frequency local details. Building on this approach, CFNO (Yang & Ren, 2023) further improved the framework by incorporating inductive lithography bias into the model architecture. While these FNO-based frameworks have demonstrated superior performance by implicitly modeling lithography in Fourier space, they did not explicitly account for the physical principles of optical diffraction theory (Dill, 1975; Hecht, 2012) that fundamentally govern lithography behavior.

To address these limitations, we propose an optical physics-inspired neural network that incorporates fundamental optical principles through the optical diffraction (Born & Wolf, 2013; Sommerfeld, 1949). In the context of semiconductor lithography, diffraction is particularly crucial as it determines how light propagates through the mask and ultimately forms patterns on silicon wafers (Moreau, 2012). A key aspect of light diffraction is the phase factor, which quantifies changes in the wavefront of light when it encounters different material interfaces (Xu et al., 2025). For instance, as light interacts with the transparent and opaque regions of a mask, the phase factor predicts how the phase and amplitude of light waves are altered during these interactions, which influence the pattern of light projected onto wafers (Erdmann et al., 2009).

To model this phase factor within our neural network, we introduce an *Optical Phase (OP) kernel* that functions as a transformation layer to encode the spatial phase variations induced by light propagation. Our framework was designed to be kernel-agnostic, allowing the use of either the Rayleigh–Sommerfeld-based diffraction kernels or the Hopkins diffraction kernel, as well as any other kernels needed. Unlike traditional convolution filters that focus only on spatial features, our kernel explicitly accounts for phase variations induced by light propagation. By incorporating this physics-inspired operation into our neural network architecture, we enhance its ability to generalize to out-of-distribution (OOD) mask designs. This capability is crucial for real-world semiconductor manufacturing, where mask patterns often deviate from the training data due to practical constraints. In addition, we adopt a simple Total Variation (TV) loss to suppress artifacts and smooth mask boundaries, further improving the manufacturability of the resulting masks.

The main contributions of our work can be summarized as follows:

- We propose a **Opti**cal diffraction-based **Co**nvolutional neural network (**OptiCo**) designed to effectively simulate the behavior of light for computational lithography.

- To the best of our knowledge, OptiCo is the first convolutional neural network framework to explicitly integrate the physical principles of optical diffraction into its core operations, thereby ensuring that the model inherently respects the underlying physics that govern lithography behavior.

- We introduce a *optical phase (OP) kernel* that encodes spatial phase variations caused by light propagation. This physics-inspired OP kernel significantly improves the generalization performance for OOD mask designs, while the addition of a TV loss further enhances mask manufacturability.

## 2 PRELIMINARIES

**Semiconductor lithography.** The primary goal of semiconductor lithography is to accurately model the interaction between mask patterns and wafers through optical projection systems. The mask $M(x, y)$ can be defined as a binary function representing opaque and transparent regions, where $(x, y)$ denote spatial coordinates on the mask plane. The mask pattern is projected onto the wafer through an optical projection framework $H$ (Hopkins, 1951; Volkmann, 1966), producing an aerial image $I(x', y') = H \cdot M(x, y)$ that represents the light intensity distribution. Here, $(x', y')$ denotes spatial coordinates on the wafer.

The final wafer pattern is determined by the interaction between the aerial image $I$ and the photoresist material on the wafer. This interaction is commonly modeled using a variable threshold resist (VTR) model (Randall et al., 1999), which captures how light intensity translates into physical features after development. In the VTR model, a dose function $D(x', y') = I(x', y') \otimes G$ is computed, where $G$ represents a spatial kernel encoding the effects of photoresist properties and exposure variability. The resulting resist image $R$ is then obtained as a binary map by applying a dose threshold $D_{\text{th}}$, such as:

$$R(x', y') = \begin{cases} 1, & \text{if } D(x', y') \geq D_{\text{th}}, \\ 0, & \text{if } D(x', y') < D_{\text{th}}. \end{cases} \tag{1}$$

This overall process from a mask $M$ to the resist image $R$ is represented as a lithography simulator $g$.

**Problem formulation.** The *mask optimization* task aims to find an optimal mask $M^*$ that produces a resist image that closely matches a desired target wafer pattern $R^*$. This task can be expressed as:

$$M^* = \arg\min_M \mathcal{L}\left(g(M), R^*\right), \tag{2}$$

where $\mathcal{L}$ is the loss function measuring the difference between the simulated resist image $R = g(M)$ and the target pattern. Conversely, the *lithography simulation* task focuses on predicting the resist image from a given mask. Accurate simulation is essential not only for validating designs but also for enabling efficient mask optimization, as it serves as the forward model in the optimization loop.

**Physics-inspired lithography models.** Early DL-based approaches to semiconductor lithography (Yang et al., 2018; Chen et al., 2020; Ye et al., 2019) primarily relied on architectures such as UNet++ (Zhou et al., 2018) and conditional GANs (Wang et al., 2018), focusing purely on data-driven learning without incorporating domain-specific physics. However, recent advancements have begun to integrate physics-inspired priors to improve model performance and generalizability. For example, DOINN (Yang et al., 2022a) and CFNO (Yang & Ren, 2023) leverage FNOs (Li et al., 2020) to implicitly encode optical physics in the frequency domain, demonstrating strong results across diverse settings. ILILT (Yang & Ren, 2024) takes this further by embedding Inverse Lithography Technology (ILT) (Zhu et al., 2023)—a physics-based framework for mask optimization—directly into the learning process. Nitho (Chen et al., 2023) introduces a coordinate-based complex multilayer perceptron (MLP) inspired by optical kernel regression (Cobb, 1995), specifically designed for lithography simulation. Although these methods embed physics-inspired principles, they rely on indirect mechanisms such as frequency-domain modeling or complex MLP. In contrast, our method explicitly incorporates physical principles into both the architectural design and the core computational operations, leading to improved generalization and robustness to OOD sets. Due to page constraints, the extended related work section is provided in Appendix F.

## 3 METHOD

### 3.1 CONVOLUTION OPERATION FROM OPTICAL DIFFRACTION

In wave optics, the propagation of an optical wave $U$ from the mask plane $(x, y)$ to a wafer plane $(x', y')$ at distance $z$ can be described by the Rayleigh-Sommerfeld (RS) diffraction integral (Sommerfeld, 1949), where $\lambda$ denotes the wavelength and $k = 2\pi/\lambda$ is the wavenumber.

The RS integral admits several forms, including the Helmholtz–Kirchhoff, the Green's function, as well as the Fresnel approximation. For clarity of presentation, we illustrate the Fresnel approximation form here, while our framework accommodates all formulations (Appendix D for further details).

$$U(x', y') = \frac{e^{jkz}}{j\lambda z} \iint_{-\infty}^{\infty} U(x, y) e^{\frac{jk}{2z}[(x'-x)^2 + (y'-y)^2]} dx dy. \tag{3}$$

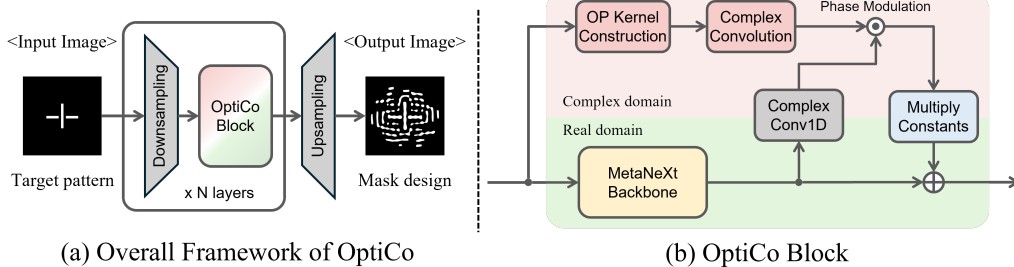

(a) Overall Framework of OptiCo        (b) OptiCo Block

Figure 2: Overview of the OptiCo framework and the design of its block integrating OP kernel.

This integral reveals that the diffraction can be modeled as a convolution with an optical propagation kernel. Recall that the convolution of a function $f$ with a kernel $h$ is mathematically defined as:

$$(f * h)(x', y') = \iint_{-\infty}^{\infty} f(x, y) h(x' - x, y' - y) \, dx \, dy. \tag{4}$$

The diffraction integral in Eq. 3 can be rewritten by isolating the quadratic dependence on $(x' - x)$ and $(y' - y)$. This transforms the integral using convolution as follows:

$$U(x', y') = \frac{e^{jkz}}{j\lambda z} \left[ U(x, y) * h(x', y') \right], \tag{5}$$

where $h(x', y')$ defines the Optical Phase (OP) kernel that characterizes diffraction through phase modulation. Note that this OP kernel appears in various formulations depending on RS forms used:

$$h(x, y) = \exp\left( \frac{jk}{2z} \left[ x^2 + y^2 \right] \right); \qquad \text{(Fresnel approximation form)}$$

$$h(x, y) = \frac{\exp\left( jk\sqrt{x^2 + y^2 + z^2} \right)}{x^2 + y^2 + z^2} \left( 1 + \frac{j}{k\sqrt{x^2 + y^2 + z^2}} \right); \quad \text{(Helmholtz–Kirchhoff form)}$$

$$h(x, y) = \frac{\exp\left( jk\sqrt{x^2 + y^2 + z^2} \right)}{x^2 + y^2 + z^2}. \qquad \text{(Green's function form)}$$

### 3.2 OPTICAL PHASE KERNEL-BASED COMPLEX CONVOLUTION

Building upon the foundation established in Section 3.1, where we demonstrated that the diffraction integral can be expressed as a convolution operation with the OP kernel, we introduce a complex convolution layer that incorporates this physically grounded kernel. Unlike traditional convolution filters that focus only on spatial features and rely entirely on data-driven learning, our formulation integrates optical priors, which enables the layer to more accurately model wave propagation and thereby enhance computational lithography performance.

**OP kernel construction.** For a kernel size of $(N, N)$, we define the coordinates $(x, y)$ relative to the center of the kernel and compute the complex exponential to construct the OP kernel $Q(x, y) = \exp\left( \frac{jk}{2z} \left[ x^2 + y^2 \right] \right)$. Each element of $Q(x, y)$ corresponds to the phase term of diffraction. In this work we mainly use the OP kernel in Fresnel form due to its effectiveness, but our framework is kernel-agnostic and can also incorporate other alternative RS formulations as well as the Hopkins transmission cross-coefficient (TCC) kernel. The pseudo-code for our OP kernel is in Appendix B.

**OP complex convolution.** To integrate the diffraction-inspired OP kernel $Q$ into our architecture, we can construct an effective kernel $W_{\text{eff}}$ in two possible forms: either element-wise multiplication with a learnable complex weight $W$, or a stricter variant scaled only by a learnable scalar weight $\lambda_\alpha$:

$$W_{\text{eff}} = Q \odot W \quad \text{or} \quad W_{\text{eff}} = \lambda_\alpha \cdot Q, \tag{6}$$

where $\odot$ denotes element-wise multiplication. The resulting complex effective kernel $W_{\text{eff}}$ encodes optical diffraction, so that each convolution step inherently reflects the physical propagation of light.

Since optical fields are inherently complex, we decompose the input $U$ and effective kernel $W_{\text{eff}}$ into real ($r$) and imaginary ($i$) components. The OP complex convolution is defined as:

$$U = U_r + jU_i; \quad W_{\text{eff}} = W_r + jW_i; \tag{7}$$

$$\text{OP}_{\text{conv}}(U) = \Big[(U_r * W_r - U_i * W_i) + j\,(U_r * W_i + U_i * W_r)\Big]. \tag{8}$$

### 3.3 Architectural Design of the OptiCo Block

The OptiCo block integrates a MetaNeXt-based backbone (Yu et al., 2024) with the OP complex convolution in a unified layer. Our backbone block takes input U as:

$$Y_{\text{backbone}}(U) = \Big(\text{DWConv}(\text{Norm}(U_r)W_1) \odot \sigma(\text{Norm}(U_r)W_2)\Big)W_3 + U_r, \tag{9}$$

where DWConv is a depthwise convolution, $\sigma$ an activation function, and $W_1, W_2, W_3$ are the MLP parameters. Since the OP convolution is inherently complex, we first embed $Y_{\text{backbone}}$ into the complex domain using a per-pixel ComplexConv1D, yielding a complex feature. At each spatial location $(p, q)$, the channel vector $x_{pq} \in \mathbb{R}^C$ is embedded into the complex domain through projections:

$$\text{ComplexConv1D}(x_{pq}) = (x_{pq,r}V_r - x_{pq,i}V_i) + j(x_{pq,r}V_i + x_{pq,i}V_r), \tag{10}$$

where $V_r, V_i \in \mathbb{R}^{C \times C}$ are learnable projection matrices. This operation serves as a per-pixel channel embedding, producing complex-valued features that can interact with the OP kernel.

This embedding is combined with $\text{OP}_{\text{conv}}$ via an inner product that performs phase modulation. The phase modulated output $Y_{\text{phase}}$ is multiplied by the constants and finally added back to the backbone feature, thereby injecting physically grounded diffraction effects into the mask representation.

$$Y_{\text{phase}}(U) = \Big[\text{ComplexConv1D}(Y_{\text{backbone}}(U)) \cdot \text{OP}_{\text{conv}}(U)\Big] \cdot \frac{e^{jkz}}{j\lambda z}, \tag{11}$$

$$Y_{\text{OptiCo}} = Y_{\text{backbone}}(U) + Y_{\text{phase}}(U). \tag{12}$$

### 3.4 Total Variation Loss for Unwanted Mask Artifact Suppression

To account for practical constraints in mask production, we aim to not only minimize the discrepancy between the target wafer pattern $R^*$ and the simulated resist image $g(M)$ from the optimized mask $M$, but also enforce a regularization loss term to suppress *grainy* artifacts that hinder mask manufacturability. The primary discrepancy loss is formally defined as:

$$\mathcal{L}_{\text{mse}}(M) = \|g(M) - R^*\|^2, \tag{13}$$

where $g(\cdot)$ denotes the lithography simulator. To promote spatial smoothness in the mask and suppress high-frequency noise, we introduce a standard 2D total variation (TV) loss (Zheng et al., 2021):

$$\mathcal{L}_{\text{tv}}(M) = \sum_{p,q}\Big|M_{p+1,q} - M_{p,q}\Big| + \Big|M_{p,q+1} - M_{p,q}\Big|. \tag{14}$$

This loss term encourages spatial smoothness by penalizing abrupt changes between adjacent pixels, thereby reducing unwanted *grainy* artifacts. Combining these two terms yields our final objective:

$$\mathcal{L}_{\text{final}}(M) = \mathcal{L}_{\text{mse}}(M) + \lambda_{\text{tv}}\,\mathcal{L}_{\text{tv}}(M), \tag{15}$$

where $\lambda_{\text{tv}}$ is the weighting parameter that controls the strength of the regularization.

## 4 Experiments

### 4.1 Experimental Design and Configurations

**Dataset.** To evaluate the performance of our OptiCo framework, we utilize LithoBench (Zheng et al., 2023b), which is one of the most recent and widely recognized computational lithography benchmarks. Specifically, LithoBench provides a comprehensive dataset of over 100,000 layout tiles, including both synthetic and real-world semiconductor designs. The dataset is composed of four

subtasks: MetalSet, ViaSet, StdMetal, and StdContact. Each subtask includes a variety of data types such as target images, optimized masks, aerial images, and resist (printed) images. The resolution of each image is 2048×2048 pixels, with each pixel representing a $1nm^2$ of the semiconductor layout.

**Setup.** In this work, we conducted experiments primarily focused on mask optimization, where the model receives target images (final wafer patterns) as input and generates optimized masks as output. This task is essential for ensuring that the resulting lithographic masks enable the precise transfer of circuit layouts onto silicon wafers. In addition to mask optimization, we carried out an additional experiment on lithography simulation modeling. In this task, the model takes optimized masks as input and produces the corresponding aerial and resist images as output. The primary objective of lithography simulation is to accurately predict the final wafer patterns, ensuring that the generated resist images closely align with the intended circuit layouts. This simulation can be viewed as a forward modeling approach that complements the inverse process of mask optimization. The hyperparameters and additional experimental details are provided in Appendix E.2.

**Evaluation metrics.** For the mask optimization task, we used three evaluation metrics: mean squared error (MSE), edge placement error (EPE) violations, and process variation band (PVB) area. Specifically, MSE measures the overall discrepancy between the simulated resist image and the target wafer pattern. EPE and PVB are industry-standard metrics commonly used in chip manufacturing to evaluate mask quality. EPE quantifies the deviation between the intended edge positions on the resist image and their actual positions. PVB measures the range of possible edge positions caused by manufacturing process variations. For the lithography simulation task, the performance of each method was assessed using two evaluation metrics: MSE and intersection over union (IOU). Detailed explanations of these evaluation metrics are provided in Appendix E.1.

**Competing methods.** We compared our OptiCo framework against several prominent semiconductor lithography models from the LithoBench, as well as recent specialized models. In particular, DAMO (Chen et al., 2020) is a fully data-driven model that uses a UNet++ (Zhou et al., 2018) backbone. In addition, DOINN (Yang et al., 2022a) and CFNO (Yang & Ren, 2023) incorporate optical physics priors by leveraging FNOs architectures. They transform the input image with a Fourier transform, embed it via MLP layers, and then concatenate those embedded features with local CNN outputs. Furthermore, ILILT (Yang & Ren, 2024) is a recent state-of-the-art mask optimization model that iteratively inputs simulated resist images along with target patterns, and Nitho (Chen et al., 2023) is a physics-inspired lithography simulation model based on optical kernel regression.

## 4.2 EXPERIMENTAL RESULTS

**Main experiments.** Table 1 provides a comprehensive evaluation of all methods across different subtasks in the mask optimization task. Our OptiCo framework consistently achieves superior performance across most evaluation metrics and subtasks, highlighting the effectiveness of integrating optical diffraction principles into the neural network architecture. Notably, OptiCo demonstrates exceptional performance on the StdMetal and StdContact subtasks, which are inherently more challenging and demand high generalization capabilities for OOD tasks.

**Strengths in OOD.** In the ViaSet, nearly all methods achieve an EPE of zero, indicating strong adaptation to the training dataset. However, on the OOD test set (StdContact), most competing methods exhibit a sharp increase in EPE. In contrast, our OptiCo consistently maintains a robust performance on the OOD test set. This robustness further highlights the effectiveness of incorporating optical diffraction principles, which offer physics-inspired guidance and enhance the model's generalization capability. Among competing methods, ILILT demonstrates strong performance. However, our OptiCo surpasses ILILT by explicitly incorporating physical principles via the OP kernel.

**Additional experiments.** Table 2 presents the performance evaluation for the lithography simulation. The evaluation metric $MSE_{resist}$ represents the MSE loss for the resist images, with the full results including $MSE_{aerial}$ reported in Appendix A.1. Consistent with our main experiments, our OptiCo outperforms all competing methods across subtasks, underscoring the robustness of OptiCo across diverse tasks. Furthermore, to evaluate the benefit of embedding optical physics, we performed a train **dataset-ratio ablation** in Appendix A.4, comparing physics-aware and purely data-driven methods. To investigate the compatibility of OptiCo, we integrated it with **different backbone networks**, such as DOINN and CFNO. As depicted in Appendix A.3, these results demonstrate that OptiCo can be seamlessly applied to diverse backbone architectures and consistently improves performance.

Table 1: Comparative performance on mask optimization across competing methods. Lower is better.

| Method | MetalSet | | | ViaSet | | | StdMetal (MetalSet OOD) | | | StdContact (ViaSet OOD) | | | Average | | |
|---|---|---|---|---|---|---|---|---|---|---|---|---|---|---|---|
| | MSE | PVB | EPE | MSE | PVB | EPE | MSE | PVB | EPE | MSE | PVB | EPE | MSE | PVB | EPE |
| DAMO | 32579 | **41173** | 5.4 | 5081 | 9962 | **0.0** | 16120 | **23796** | 0.2 | 50445 | 35673 | 26.7 | 26056 | 27651 | 8.1 |
| DOINN | 36409 | 41929 | 7.4 | 4382 | 7836 | **0.0** | 25913 | 25749 | 4.5 | 72058 | 17968 | 55.8 | 34691 | **23370** | 16.9 |
| CFNO | 47814 | 46131 | 12.5 | 8949 | 9890 | 0.1 | 26809 | 26814 | 4.2 | 70740 | **17950** | 55.1 | 38578 | 25196 | 18.0 |
| ILILT | 30353 | 45353 | 3.2 | 4666 | 10065 | **0.0** | 14596 | 24969 | 0.1 | 38957 | 43869 | 7.1 | 22143 | 31064 | 2.6 |
| OptiCo | **24033** | 45379 | **1.6** | **4339** | **7802** | **0.0** | **11293** | 25183 | **0.0** | **18474** | 39181 | **0.1** | **14535** | 29373 | **0.4** |

Table 2: Additional experiments on lithography simulation.

| Method | StdMetal ($MSE_{resist}$) | StdContact ($MSE_{resist}$) |
|---|---|---|
| DAMO | $1.50 \cdot 10^{-3}$ | $1.64 \cdot 10^{-3}$ |
| DOINN | $1.29 \cdot 10^{-3}$ | $1.37 \cdot 10^{-3}$ |
| CFNO | $2.29 \cdot 10^{-3}$ | $2.20 \cdot 10^{-3}$ |
| Nitho | $1.94 \cdot 10^{-3}$ | $1.82 \cdot 10^{-3}$ |
| OptiCo | $\mathbf{4.18 \cdot 10^{-4}}$ | $\mathbf{1.24 \cdot 10^{-3}}$ |

Table 3: OP kernel ablation on alternative diffraction formulas.

| Diffraction formulas | StdMetal (EPE) | StdContact (EPE) |
|---|---|---|
| w/o kernel | 2.819 | 22.612 |
| Helmholtz | 0.207 | 0.303 |
| Hopkins | 0.125 | 0.461 |
| Green's | 0.059 | 0.188 |
| Fresnel | **0.044** | **0.079** |

Table 4: Ablation on stricter OP kernel with scalar weight $\lambda_\alpha$.

| Diffraction formulas | StdMetal (EPE) | StdContact (EPE) |
|---|---|---|
| w/o kernel | 2.819 | 22.612 |
| Helmholtz | 0.786 | 5.497 |
| Hopkins | **0.044** | **0.212** |
| Green's | 0.059 | 0.309 |
| Fresnel | 0.173 | 1.061 |

### 4.3 ABLATION STUDY

**Alternative diffraction kernel.** OP kernel is a key contribution of OptiCo that embeds diffraction principles into the network. We replaced the default Fresnel OP kernel with alternative diffraction formulations. In Table 3, we found that the Helmholtz-based kernel led to lower performance, likely due to its complexity, while the Green and Fresnel kernels achieved stronger results, suggesting that their simpler form integrates more effectively into neural architectures.

**Strict OP kernel variant.** We examined a stricter effective kernel variant ($W_{\text{eff}} = \lambda_\alpha Q$) in which the OP kernel is scaled only by a scalar weight, as shown in Table 4. In this setting, the Green and Hopkins kernels outperformed the Fresnel kernel, which we attribute to their higher numerical aperture (NA) suitability compared to the paraxial Fresnel approximation.

**Kernel size ablation.** A kernel that is too small results in a limited receptive field, making it insufficient to capture a diffraction principle, while a large kernel introduces complexity. As shown in Appendix A.5, performance initially improves as the kernel size increases, but degrades when it becomes too large. We attribute this to the quadratic nature of a kernel ($x^2 + y^2$), as steep phase variations hinder training. Nevertheless, OptiCo consistently outperforms strong baselines (ILILT).

**Advantage of modeling in complex domain.** Since phase is a crucial component of light, modeling in the complex domain is necessary. To validate this, we ablated OP kernel, applying only ComplexConv1D (CC). Even this simple modification leads to a notable performance gain, confirming the effectiveness of complex-domain modeling.

**Completeness of phase modulation.** In the OptiCo block, multiplying constants (MC) are essential for preserving the full diffraction formulation. When these constants are ablated from OP convolution, we observe a substantial performance drop, highlighting the importance of physical completeness. Finally, adding ComplexConv1D makes the phase modulation complete and further boosts performance, underscoring the necessity of a physically faithful design.

Table 5: An ablation study for each component in OptiCo block: OP complex convolution (OP), Multiply Constants (MC), ComplexConv1D (CC).

| Ablation study | | | StdMetal (EPE) | StdContact (EPE) |
|---|---|---|---|---|
| OP | MC | CC | | |
| - | - | - | 2.819 | 22.612 |
| - | - | ✔ | 0.657 | 7.491 |
| ✔ | - | - | 1.561 | 6.321 |
| ✔ | ✔ | - | 0.188 | 1.273 |
| ✔ | ✔ | ✔ | **0.044** | **0.079** |

**Comparison with FNO.** OptiCo explicitly incorporates diffraction principles, whereas FNO reflects them implicitly through Fourier modeling. To examine this distinction, we replaced the OptiCo block with an FNO module and compared performance. As shown in Figure 3, substituting FNO yields a clear performance gain, consistent with our earlier finding on the importance of complex-domain modeling. Nevertheless, OptiCo achieves superior performance by explicitly embedding diffraction principles. Additional evaluation metrics are provided in Appendix A.2.

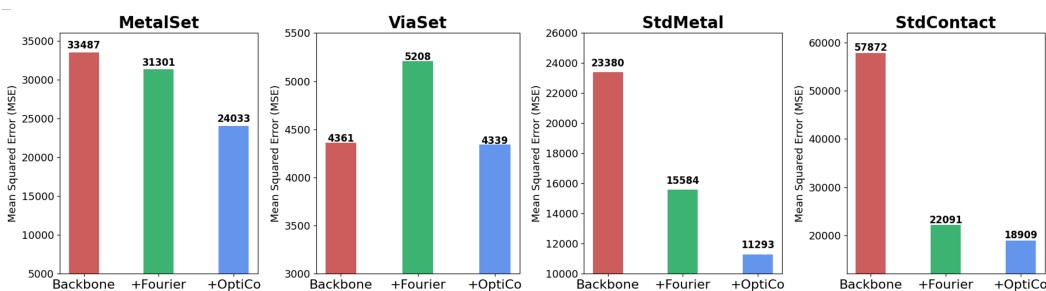

Figure 3: Comparative performance of the backbone network with the addition of Fourier module (FNO) and our OptiCo that contains the OP kernel. The results highlight the effectiveness of OptiCo.

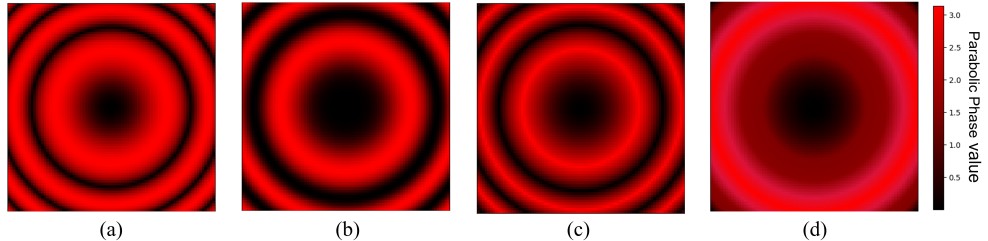

Figure 4: Visualization of the OP kernel, illustrating dynamic adjustments across wavefront scales.

### 4.4 VISUALIZATION OF DIFFRACTION-INSPIRED REPRESENTATIONS

**Visualization of the OP kernel.** To provide an intuitive understanding of how the OP kernel encodes diffraction principles, we visualize its spatial structure across different wavefront scales. As illustrated in Figure 4, the images (a), (b), (c), and (d) represent distinct configurations of the OP kernel, $\exp\left(\frac{jk}{2z}\left[x^2 + y^2\right]\right)$, each illustrating its dynamic adaptation to varying wavefront parameters. The accompanying color map on the right quantitatively represents parabolic phase values, where darker shades correspond to lower phases and brighter reds indicate higher accumulations. A key observation across all images is the presence of concentric rings with alternating intensity—an intrinsic characteristic of the parabolic phase term in diffraction. These rings indicate phase variations that reflect the OP kernel's capacity to accurately model light propagation. Comparing images (a) through (d), we observe dynamic changes in the spacing, sharpness, and intensity gradients of the rings. Specifically, in (a), the rings are tightly packed, indicating high-frequency phase variations, while in (b) and (c), the rings are more widely spaced, reflecting a lower-frequency phase structure.

**Intermediate feature visualization.** To further examine how the OP kernel influences internal representations, we provide additional intermediate feature visualizations comparing models with and without OptiCo in Appendix C.2. These results show that OptiCo focuses on diffraction-sensitive pattern corners, highlighting the model's ability to capture optical principles. In summary, these visual representations underscore the OP kernel's capacity to accurately model the complex optical lithography behavior.

### 4.5 VISUALIZATION OF DIFFRACTION-INSPIRED MASK PATTERNS

We compared the mask patterns optimized by OptiCo with those from other competing methods in Figure 5. Notably, OptiCo produces mask patterns featuring more distinct outer ring structures than its competitors. These diffraction-inspired ring structures arise from integrating light diffraction principles through the OP kernel, as shown in Figure 4. In particular, this phenomenon is especially pronounced when comparing ViaSet with its OOD test set, StdContact. OptiCo produces highly structured mask patterns in both datasets, clearly guided by the OP kernel's explicit modeling of light diffraction. In contrast, DOINN, which is an FNO-based framework, forms complex diffraction patterns on the ViaSet but fails to replicate them on the StdContact (OOD test set). This discrepancy underscores the advantage of explicitly incorporating optical diffraction principles into semiconductor lithography through the use of our OP kernel. Additional visualizations for all methods and other subtasks are provided in Appendix C.1.

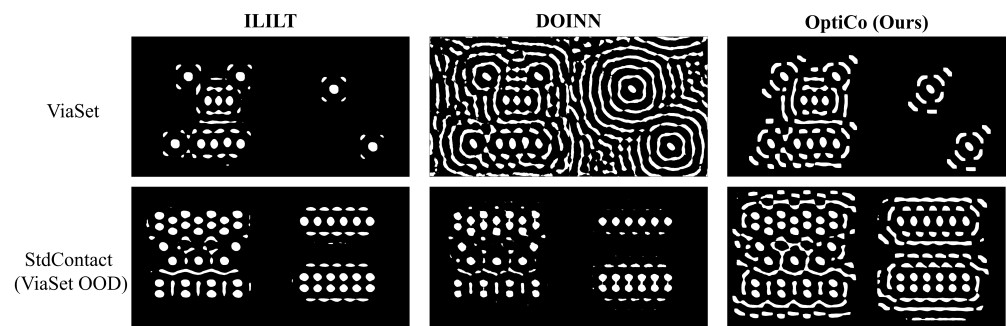

Figure 5: Visualization of optimized mask patterns. OptiCo produces clearer diffraction-inspired rings, especially on StdContact, highlighting the benefits of explicit diffraction modeling. In contrast, the Fourier-based DOINN shows complex rings on ViaSet but fails to preserve them on StdContact.

### 4.6 MANUFACTURABILITY IMPROVEMENT WITH TV LOSS

**Artifact suppression.** In mask optimization, *grainy artifacts* refer to small, high-frequency variations or speckles present in the mask pattern. While these artifacts may appear insignificant, they present substantial challenges in practical manufacturing, as they can lead to unpredictable deviations in the final pattern on the wafer. As shown in Figure 6, we compare the grainy artifacts in optimized mask patterns with and without the total variation (TV) loss. The model without TV loss produces noticeable grainy artifacts (highlighted in red), which pose practical challenges in mask manufacturing. In contrast, integrating TV loss into our model largely suppresses these grainy artifacts, resulting in smoother boundaries and more coherent shapes. This highlights the advantage of TV loss in enhancing the manufacturability of the masks.

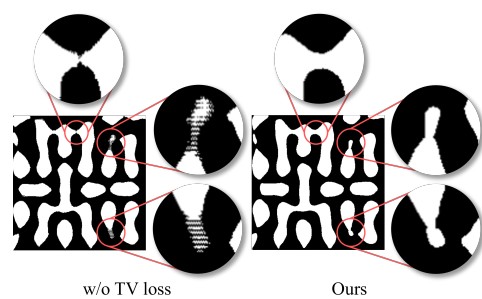

w/o TV loss          Ours

Figure 6: Comparison of mask patterns with and without the total variation (TV) loss, highlighting reduced grainy artifacts and smoother boundaries with coherent shapes in red circle.

**Simplified mask and post-processing robustness.** We also assessed manufacturability by comparing shot counts, which reflect the complexity of the layout geometry. As in Table 6, incorporating TV loss leads to a slight degradation in EPE but a substantial improvement in shot count, reducing it to less than half of that without TV loss. For further analysis, we applied morphological post-processing to the mask images (Yang & Ren, 2025), which is commonly used to suppress artifacts and smooth boundaries, though

Table 6: TV loss produces cleaner masks, leading to minimal EPE degradation and stable shot count after post-processing.

| Loss function | EPE ($\downarrow$) | Shot ($\downarrow$) |
|---|---|---|
| $\mathcal{L}_{\text{mse}}$ | 1.4 | 1523 |
| $\mathcal{L}_{\text{mse}}$ + PP | 2.0 ($\Delta$+0.6) | 1451 ($\Delta$-72) |
| $\mathcal{L}_{\text{mse}} + \mathcal{L}_{\text{tv}}$ | 1.6 | 715 |
| $\mathcal{L}_{\text{mse}} + \mathcal{L}_{\text{tv}}$ + PP | 1.8 ($\Delta$+0.2) | 712 ($\Delta$-3) |

it inevitably degrades performance. Intuitively, if the mask is clean, the performance drop after post-processing should remain small. Indeed, masks trained with TV loss exhibit only minor degradation in EPE and negligible changes in shot count, confirming that TV loss promotes cleaner masks.

## 5 CONCLUSION

In this paper, we propose OptiCo, a novel physics-inspired convolutional neural network that integrates diffraction principles directly into its architecture to enhance computational lithography. Unlike conventional approaches that rely solely on data-driven methods or Fourier-based frameworks, OptiCo introduces a physics-inspired OP kernel to explicitly model the near-field diffraction behavior of light. Empirical evaluations on well-established lithography benchmarks demonstrate that our OptiCo consistently outperforms existing state-of-the-art methods in both lithography simulation and mask optimization tasks. Moreover, our in-depth analysis underscores the crucial role of the OP kernel in explicitly capturing the parabolic phase variations inherent to diffraction principles.

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

# A  ADDITIONAL EXPERIMENTS AND RESULTS

## A.1  FULL RESULTS ON MASK OPTIMIZATION AND LITHOGRAPHY SIMULATION

Table 7: Comparative performance on (a) mask optimization, evaluated using MSE, PVB, and EPE, and (b) lithography simulation, assessed via MSE (aerial/resist) and IOU. Best results are in bold.

| (a) Mask optimization | | | | | (b) Lithography simulation | | | | |
|---|---|---|---|---|---|---|---|---|---|
| **Subtask** | **Method** | **MSE↓** | **PVB↓** | **EPE↓** | **Subtask** | **Method** | **MSE$_{aerial}$↓** | **MSE$_{resist}$↓** | **IOU↑** |
| MetalSet | DAMO | 32579 | **41173** | 5.4 | MetalSet | DAMO | $8.36 \cdot 10^{-6}$ | $7.48 \cdot 10^{-4}$ | 0.97 |
| | DOINN | 36409 | 41929 | 7.4 | | DOINN | $9.45 \cdot 10^{-6}$ | $6.77 \cdot 10^{-4}$ | 0.97 |
| | CFNO | 47814 | 46131 | 12.5 | | CFNO | $1.87 \cdot 10^{-5}$ | $1.47 \cdot 10^{-3}$ | 0.94 |
| | ILILT | 30353 | 45353 | 3.2 | | Nitho | $9.27 \cdot 10^{-6}$ | $7.95 \cdot 10^{-4}$ | 0.97 |
| | OptiCo | **24033** | 45327 | **1.6** | | OptiCo | $\mathbf{2.44 \cdot 10^{-6}}$ | $\mathbf{4.18 \cdot 10^{-4}}$ | **0.98** |
| ViaSet | DAMO | 5081 | 9962 | **0.0** | ViaSet | DAMO | $2.98 \cdot 10^{-6}$ | $1.47 \cdot 10^{-4}$ | 0.94 |
| | DOINN | 4382 | 7836 | **0.0** | | DOINN | $2.02 \cdot 10^{-6}$ | $1.04 \cdot 10^{-4}$ | **0.96** |
| | CFNO | 8949 | 9890 | 0.1 | | CFNO | $3.77 \cdot 10^{-6}$ | $2.13 \cdot 10^{-4}$ | 0.92 |
| | ILILT | 4666 | 10065 | **0.0** | | Nitho | $2.24 \cdot 10^{-6}$ | $1.46 \cdot 10^{-4}$ | 0.95 |
| | OptiCo | **4339** | **7802** | **0.0** | | OptiCo | $\mathbf{1.04 \cdot 10^{-6}}$ | $\mathbf{8.88 \cdot 10^{-5}}$ | **0.96** |
| StdMetal (MetalSet OOD) | DAMO | 16120 | **23796** | 0.2 | StdMetal (MetalSet OOD) | DAMO | $2.50 \cdot 10^{-5}$ | $1.50 \cdot 10^{-3}$ | 0.96 |
| | DOINN | 25913 | 25749 | 4.5 | | DOINN | $1.36 \cdot 10^{-5}$ | $1.29 \cdot 10^{-3}$ | 0.96 |
| | CFNO | 26809 | 26814 | 4.2 | | CFNO | $2.61 \cdot 10^{-5}$ | $2.29 \cdot 10^{-3}$ | 0.93 |
| | ILILT | 14596 | 24969 | 0.1 | | Nitho | $2.80 \cdot 10^{-5}$ | $1.94 \cdot 10^{-3}$ | 0.95 |
| | OptiCo | **11293** | 25183 | **0.0** | | OptiCo | $\mathbf{2.44 \cdot 10^{-6}}$ | $\mathbf{4.18 \cdot 10^{-4}}$ | **0.98** |
| StdContact (ViaSet OOD) | DAMO | 50445 | 35673 | 26.7 | StdContact (ViaSet OOD) | DAMO | $4.56 \cdot 10^{-5}$ | $1.64 \cdot 10^{-3}$ | 0.87 |
| | DOINN | 72058 | 17968 | 55.8 | | DOINN | $2.72 \cdot 10^{-5}$ | $1.37 \cdot 10^{-3}$ | 0.89 |
| | CFNO | 70740 | **17950** | 55.1 | | CFNO | $2.14 \cdot 10^{-5}$ | $2.20 \cdot 10^{-3}$ | 0.83 |
| | ILILT | 38957 | 43869 | 7.1 | | Nitho | $2.22 \cdot 10^{-5}$ | $1.82 \cdot 10^{-3}$ | 0.85 |
| | OptiCo | **18909** | 39181 | **0.1** | | OptiCo | $\mathbf{1.01 \cdot 10^{-5}}$ | $\mathbf{1.24 \cdot 10^{-3}}$ | **0.90** |
| Average | DAMO | 26056 | 27651 | 8.1 | Average | DAMO | $2.05 \cdot 10^{-5}$ | $1.01 \cdot 10^{-3}$ | 0.93 |
| | DOINN | 34691 | **23370** | 16.9 | | DOINN | $1.31 \cdot 10^{-5}$ | $8.61 \cdot 10^{-4}$ | 0.95 |
| | CFNO | 38578 | 25196 | 18.0 | | CFNO | $1.75 \cdot 10^{-5}$ | $1.54 \cdot 10^{-3}$ | 0.90 |
| | ILILT | 22143 | 31064 | 2.6 | | Nitho | $1.54 \cdot 10^{-5}$ | $1.18 \cdot 10^{-3}$ | 0.93 |
| | OptiCo | **14535** | 29373 | **0.4** | | OptiCo | $\mathbf{4.00 \cdot 10^{-6}}$ | $\mathbf{5.40 \cdot 10^{-4}}$ | **0.96** |

Table 7 reports the comprehensive performance evaluation on both mask optimization and lithography simulation tasks. For the lithography simulation task, we provide full results with the evaluation metrics MSEaerial, MSEresist, and IoU. Here, MSEaerial and MSEresist measure the mean squared error of the aerial and resist images, respectively, while IoU quantifies the overlap between the predicted and target resist patterns. Consistent with the main paper, our proposed OptiCo consistently outperforms all competing methods across subtasks, further underscoring its robustness and generality.

Among the competing approaches, FNO-based methods such as DOINN and CFNO achieve stronger performance than DAMO, which is fully data-driven yet computationally expensive. This trend highlights the advantage of incorporating Fourier-based operators, which effectively capture both low-frequency global information and high-frequency local details within Fourier space.

It may be noted that the Hopkins TCC kernel–inspired baseline Nitho exhibits relatively lower performance in our setting. We attribute this performance drop primarily to its lightweight architectural design, which emphasizes efficiency with fewer parameters and reduced training data. While our OptiCo takes a different approach and achieves stronger accuracy in several scenarios, Nitho remains a well-designed and meaningful model with a distinct focus on efficiency for rapid lithography simulation.

## A.2    ADDITIONAL EVALUATION METRICS FOR FOURIER MODULE COMPARISON

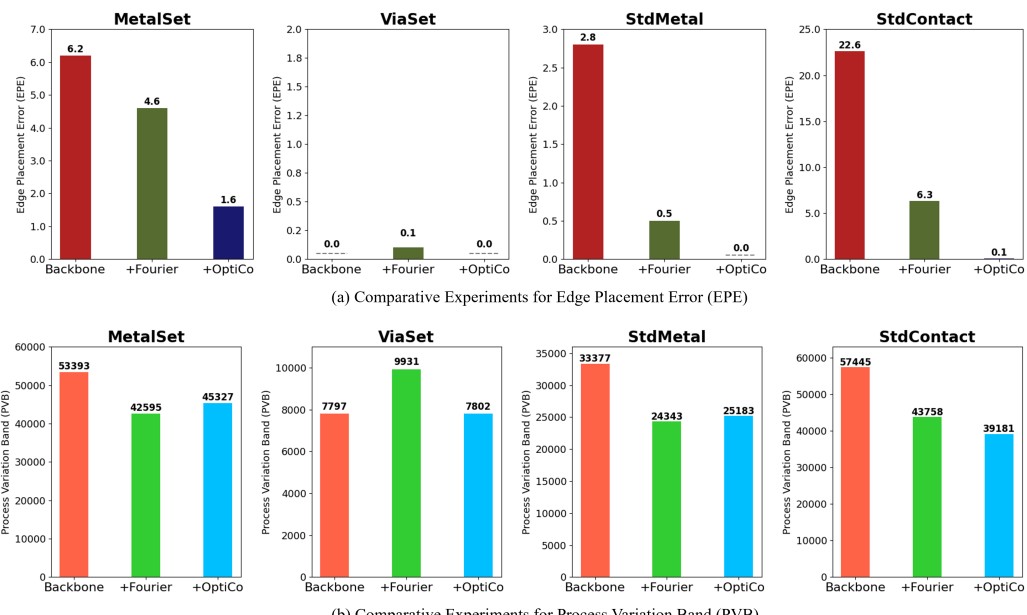

(a) Comparative Experiments for Edge Placement Error (EPE)

(b) Comparative Experiments for Process Variation Band (PVB)

Figure 7: Comparative performance of the backbone network with the addition of the Fourier module (FNO) and our OptiCo module that contains the OP kernel. The results highlight the superior performance of our OptiCo module with the OP kernel in reducing EPE and PVB.

In the main manuscript, we conducted comparative experiments across three configurations: 1) a vanilla backbone network, 2) a backbone network with a Fourier module, and 3) a backbone network integrated with our OptiCo module, which incorporates the OP kernel, evaluated using the MSE metric. In this section, we extend the comparative experiments to other evaluation metrics, such as Edge Placement Error (EPE) and Process Variation Band (PVB).

As illustrated in Figure 7 (a), our OptiCo framework demonstrated exceptional performance in the EPE metric as well. This was particularly evident on OOD datasets such as StdMetal and StdContact, where OptiCo exhibited remarkable improvements. These results highlight the effectiveness of integrating physics-inspired Fresnel diffraction principles into the neural network architecture, allowing OptiCo to accurately model light propagation and diffraction effects.

Similarly, Figure 7 (b) presents the results for the PVB metric, where OptiCo consistently delivered superior performance, particularly on OOD datasets. While the Fourier-based module outperformed the vanilla backbone network—highlighting the advantages of operating within the Fourier space to implicitly model global light propagation patterns—our OptiCo module explicitly modeled these patterns through the OP kernel. This explicit modeling approach enabled OptiCo to achieve superior performance, clearly demonstrating the advantages of explicitly capturing the phase variations inherent in diffraction over implicit modeling approaches.

These additional results for comparative performance collectively validate the robustness and generalization capabilities of our OptiCo framework across diverse evaluation metrics and challenging OOD datasets.

## A.3 OptiCo with Different Backbone Networks

Table 8: Performance comparison of OptiCo framework integrated with various backbone networks such as DOINN and CFNO.

| Subtask | Method | MSE(↓) | PVB(↓) | EPE(↓) |
|---------|--------|--------|--------|--------|
| MetalSet | DOINN | 36409 | **41929** | 7.4 |
| | + OptiCo (ours) | **28322** | 44846 | **3.1** |
| | CFNO | 47814 | 46131 | 12.5 |
| | + OptiCo (ours) | **38586** | **45743** | **7.5** |
| ViaSet | DOINN | 4382 | 7836 | **0.0** |
| | + OptiCo (ours) | **4362** | **7819** | **0.0** |
| | CFNO | 8949 | 9890 | 0.1 |
| | + OptiCo (ours) | **4406** | **8505** | **0.0** |
| StdMetal (MetalSet OOD) | DOINN | 25913 | 25749 | 4.5 |
| | + OptiCo (ours) | **15589** | **25408** | **0.8** |
| | CFNO | 26809 | 26814 | 4.2 |
| | + OptiCo (ours) | **22542** | **25328** | **2.5** |
| StdContact (ViaSet OOD) | DOINN | 72058 | **17968** | 55.8 |
| | + OptiCo (ours) | **23213** | 40113 | **2.2** |
| | CFNO | 70740 | **17950** | 55.1 |
| | + OptiCo (ours) | **23150** | 42731 | **1.7** |
| Average | DOINN | 34691 | **23370** | 16.9 |
| | + OptiCo (ours) | **17871** | 29546 | **1.5** |
| | CFNO | 38578 | **25196** | 18.0 |
| | + OptiCo (ours) | **22171** | 30577 | **2.9** |

In our main experiments, we employed a MetaNeXt-based backbone neural network within the OptiCo framework to demonstrate its efficacy in lithography simulation and mask optimization tasks. To further validate the versatility and compatibility of OptiCo, we extended its integration to a range of other FNO-based backbone networks, specifically DOINN and CFNO, which also serve as competing methods in our study.

As detailed in Table 8, our OptiCo framework was systematically integrated with both DOINN and CFNO backbones. The results demonstrate that OptiCo seamlessly integrates with diverse backbone architectures and consistently enhances performance across most evaluation metrics. Notably, its incorporation with the DOINN and CFNO backbones yielded significant improvements on challenging OOD datasets such as StdMetal and StdContact.

These experiments highlight that OptiCo is not limited to a single backbone architecture but can be seamlessly applied to various backbone networks. The consistent performance enhancements observed across different backbones indicate that the benefits of incorporating the Optical Phase (OP) kernel are universally applicable, regardless of the underlying neural network architecture. This adaptability is particularly valuable in computational lithography, where different backbone networks may be preferred based on specific task requirements or computational constraints.

## A.4 Dataset Ablation

Physics-based inductive biases are known to substantially improve data efficiency (Rao et al., 2022). To quantify this effect, we conducted a dataset ablation study varying the training dataset size (10%, 25%, 50%, and 100%). Specifically, we compared two physics-aware models—ILILT and our OptiCo model—with their purely data-driven counterparts—DAMO and a variant of OptiCo without the OP kernel. As shown in Table 9 and Figure 8, models incorporating physics priors achieve significantly lower EPE scores at all data regimes, demonstrating their superior efficiency under limited data.

Table 9: Average EPE comparisons from dataset ablation study (lower is better). Models with physics-based priors (ILILT and OptiCo) show significantly better data efficiency under limited data.

| Training ratio | Without physics prior | | With physics prior | |
|---|---|---|---|---|
| | DAMO | Ours w/o OP | ILILT | Ours |
| 10% | 73.2 | 45.8 | 7.3 | 6.4 |
| 25% | 35.0 | 17.3 | 5.5 | 1.1 |
| 50% | 12.9 | 11.9 | 4.5 | 0.8 |
| 100% | 8.1 | 8.1 | 2.6 | 0.4 |

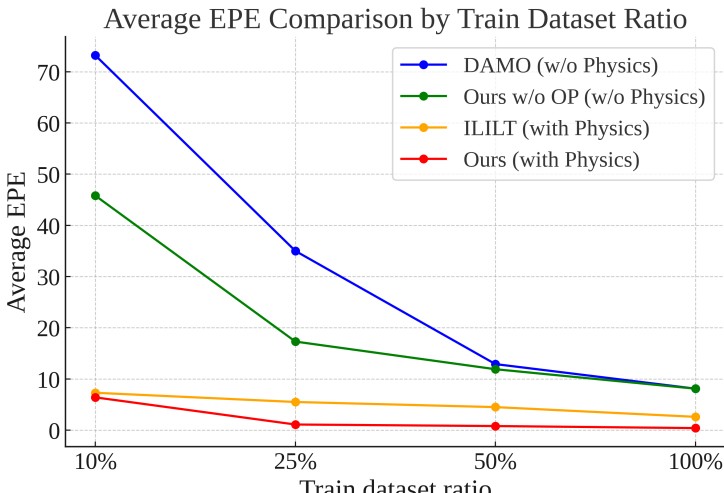

Figure 8: Ablation studies of training dataset ratio. We compare the performance of models with and without physics priors across varying ratios of training data.

## A.5 KERNEL SIZE ABLATION

Since OptiCo is a convolution-based framework, the OP kernel size is a critical factor in its performance. A kernel that is too small results in a limited receptive field, making it insufficient to capture diffraction. We conduct an ablation study on the OP kernel size in Table 3. Increasing the kernel size initially improves performance, but beyond a certain point, it begins to degrade, indicating that the performance does not improve monotonically. We attribute this trend to the quadratic nature of the OP, $(x^2 + y^2)$. When the kernel becomes too large, steep phase variations may disrupt effective learning. Nevertheless, OptiCo consistently outperforms the strong baseline ILILT across all tested kernel sizes, highlighting the robustness.

Table 10: Ablation on the kernel size.

| Kernel size | StdMetal (EPE↓) | StdContact (EPE↓) |
|---|---|---|
| 3 | 0.048 | 0.364 |
| 5 | 0.048 | 0.345 |
| 7 | 0.055 | 0.139 |
| 9 | 0.052 | 0.097 |
| 11 | **0.044** | **0.079** |
| 13 | 0.055 | 0.182 |
| 15 | 0.050 | 0.221 |
| 17 | 0.048 | 0.242 |
| ILILT | 0.111 | 7.055 |

## B PYTORCH-LIKE ALGORITHM FOR OP KERNEL CONSTRUCTION

For clarity, we provide a PyTorch-style algorithm for constructing the OP kernel used in our complex convolution layer in Section 3.2. This code snippet directly reflects the OP formulation for Fresnel approximation form, where the kernel encodes the parabolic phase term derived from the Fresnel diffraction. Each element of the constructed OP kernel corresponds to the complex exponential, $Q(x, y) = \exp\left(\frac{jk}{2z}(x^2 + y^2)\right)$.

```python
def create_Q_kernel(k, z, kernel_size):
    N = kernel_size
    center = (N - 1)/2
    x = torch.arange(N) - center
    y = torch.arange(N) - center
    X, Y = torch.meshgrid(x, y, index='ij')

    exponent = 1j*k*(X**2 + Y**2)/(2*z)
    Q = torch.exp(exponent)
    return Q
```

Algorithm 1: PyTorch implementation of the OP kernel

## C ADDITIONAL VISUALIZATION OF MASK PATTERNS AND INTERMEDIATE FEATURES

### C.1 ADDITIONAL SAMPLE MASK PATTERN VISUALIZATIONS

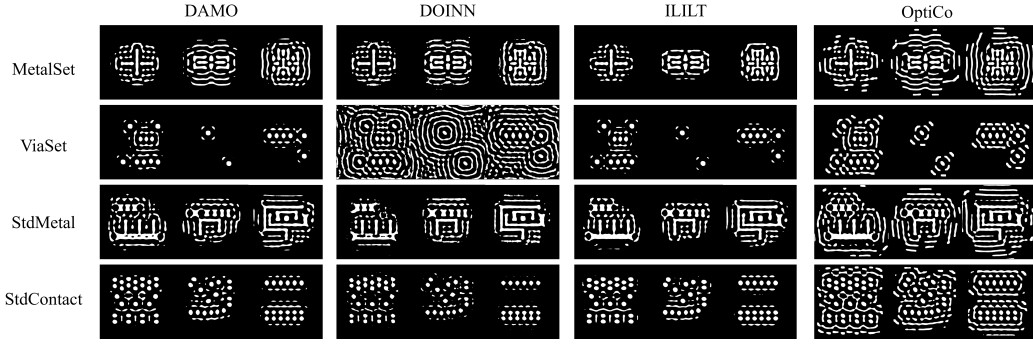

Figure 9: Visualization of additional optimized mask patterns across all competing methods.

In our main manuscript, we presented visualizations of sample mask patterns for strong competing methods such as DOINN and ILILT. To complement the main manuscript, Figure 9 presents a more comprehensive set of visualizations to allow for a broader and more detailed comparison across other competing methods. These additional visualizations allow for a more thorough comparison and highlight the distinctive advantages of our approach.

As previously discussed, it is evident from the visualizations that our OptiCo consistently generates mask patterns with more distinct outer ring structures compared to other methods. This clarity in the outer rings underscores the effectiveness of integrating light diffraction principles through the *Optical Phase (OP) kernel* within our neural network architecture. The pronounced ring structures indicate accurate modeling of spatial phase variations, which are crucial for precisely capturing lithographic behavior in computational lithography. In contrast, other competing methods struggle to generate well-defined mask patterns, particularly on OOD datasets such as StdMetal and StdContact. This performance degradation can be attributed to their reliance on implicit modeling approaches, which lack the explicit phase-aware transformations provided by OptiCo.

## C.2 Intermediate Feature Visualization

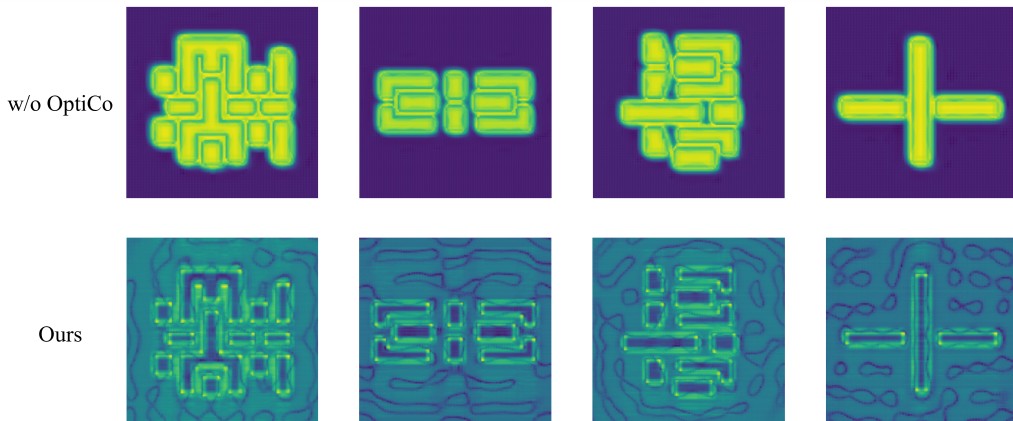

w/o OptiCo

Ours

Figure 10: Visualization of intermediate features with and without the physics-informed OptiCo block. The intermediate features produced by OptiCo explicitly emphasize critical corner regions highlighted by bright features around them. In contrast, intermediate features from the baseline method focus only on the mask regions, neglecting the critical corners.

To better understand how the OP kernel influences OptiCo's internal representations, we visualize intermediate feature maps from models with and without OptiCo in Figure 10. Pattern corners are key regions in lithography due to their sensitivity to light diffraction, making them important targets for effective modeling. We observe that OptiCo consistently emphasizes corner regions and nearby sub-resolution assist features (SRAFs, see Appendix F.2 for details) through strong activations. In contrast, the baseline model without OptiCo tends to highlight only the main mask shapes, failing to distinguish critical corners from other regions and largely ignoring surrounding diffraction-sensitive areas. These results suggest that the OP kernel encourages the network to focus on phase-sensitive areas such as corners, thereby enhancing the model's understanding of the optical lithography process.

## D Diffraction Integrals

In scalar diffraction theory, the unknown field $U$ is obtained by representing it via the Green's function and a boundary integral. The Rayleigh–Sommerfeld (RS) diffraction integral serves as the standard starting point for exact diffraction calculations, and controlled approximations yield simpler and computationally convenient expressions.

**Helmholtz–Kirchhoff form.** Derived from the Helmholtz–Kirchhoff integral theorem with Kirchhoff boundary conditions, it expresses the field as an aperture integral and retains the obliquity factor $z/r$ (angular dependence), it also keeps the near-field correction $1 + j/kr$. Accordingly, it is the RS representation with the fewest approximations. Let $r = \sqrt{(x'-x)^2 + (y'-y)^2 + z^2}$,

$$U(x',y') = \frac{1}{j\lambda} \iint_{-\infty}^{\infty} U(x,y)\frac{e^{jkr}}{r}\frac{z}{r}\left(1 + \frac{j}{kr}\right) dxdy$$

$$U(x',y') = \frac{z}{j\lambda} \iint_{-\infty}^{\infty} U(x,y)\frac{e^{jkr}}{r^2}\left(1 + \frac{j}{kr}\right) dxdy$$

$$U(x',y') = \frac{z}{j\lambda} \cdot U(x,y) * \left[\frac{\exp\left(jk\sqrt{x^2+y^2+z^2}\right)}{x^2+y^2+z^2}\left(1 + \frac{j}{k\sqrt{x^2+y^2+z^2}}\right)\right] \tag{16}$$

$$h(x,y) = \frac{\exp\left(jk\sqrt{x^2+y^2+z^2}\right)}{x^2+y^2+z^2}\left(1 + \frac{j}{k\sqrt{x^2+y^2+z^2}}\right)$$

As you can see from the form of the equation, the Helmholtz–Kirchhoff kernel is relatively complex, which makes it less practical when used as a convolutional kernel in a neural network. This observation is consistent with our ablation study in Table 3, where the Helmholtz-based kernel shows significantly degraded performance compared to the alternatives.

**Green's function form.** The Green's-function form is the canonical Rayleigh–Sommerfeld diffraction integral. In our convention, it is derived directly from the free-space Green's function for the scalar Helmholtz equation, without explicitly invoking the Kirchhoff boundary condition on the opaque screen. Modeling the aperture as a planar boundary distribution and enforcing the Sommerfeld radiation condition omits the near-field correction $(1 + j/kr)$, resulting in a simplified but widely used expression. This form is frequently adopted in practice due to its reduced complexity while still capturing the essential diffraction behavior.

$$U(x', y') = \frac{1}{j\lambda} \iint_{-\infty}^{\infty} U(x, y) \frac{e^{jkr}}{r} \frac{z}{r} dxdy$$

$$U(x', y') = \frac{z}{j\lambda} \iint_{-\infty}^{\infty} U(x, y) \frac{e^{jkr}}{r^2} dxdy$$

$$U(x', y') = \frac{z}{j\lambda} \cdot U(x, y) * \left[ \frac{\exp\left(jk\sqrt{x^2 + y^2 + z^2}\right)}{x^2 + y^2 + z^2} \right] \tag{17}$$

$$h(x, y) = \frac{\exp\left(jk\sqrt{x^2 + y^2 + z^2}\right)}{x^2 + y^2 + z^2}$$

This form is widely employed due to its reduced complexity while still providing an accurate description of diffraction. As shown in Tables 3 and 4, the Green's function form consistently performs well, underscoring its practical value and motivating further study.

**Fresnel approximation form.** The Fresnel approximation employed in the main text is one of two key approximations to the Rayleigh-Sommerfeld integral under near-field conditions. The primary objective of the Fresnel approximation is to approximate the distance $r$, which assumes that distance $z$ is sufficiently large compared to the aperture size of the mask. Under this condition, the oblique factor $z/r$ can be approximated to 1, as known as paraxial approximation.

$$\frac{z}{r} \approx 1 \tag{18}$$

Here, $r$ is approximated via a binomial expansion, known as the Fresnel approximation.

$$r \approx z + \frac{(x - x')^2 + (y - y')^2}{2z}. \tag{19}$$

Starting with Green's function form, we substitute $r$ and oblique factor $z/r$ from the equation.

$$U(x', y') = \frac{1}{j\lambda} \iint_{-\infty}^{\infty} U(x, y) \frac{e^{jkr}}{r} \frac{z}{r} dxdy$$

$$U(x', y') = \frac{e^{jkz}}{j\lambda z} \iint_{-\infty}^{\infty} U(x, y) e^{\frac{jk}{2z}[(x'-x)^2 + (y'-y)^2]} dxdy.$$

$$U(x', y') = \frac{e^{jkz}}{j\lambda z} \left[ U(x, y) * \exp\left(\frac{jk}{2z}\left[x^2 + y^2\right]\right) \right], \tag{20}$$

$$h(x, y) = \exp\left(\frac{jk}{2z}\left[x^2 + y^2\right]\right)$$

We adopt this kernel as our primary optical-phase (OP) kernel. Its simplicity enables accurate and efficient diffraction modeling.

**Hopkins transmission cross-coefficient (TCC) model.** The Hopkins model is a widely used framework in computational lithography to simulate the optical imaging process. The Hopkins model calculates the aerial image $I(x', y')$, representing the light intensity at the wafer plane, as follows:

$$I(x', y') = \iiiint_{-\infty}^{\infty} \mathcal{F}(M)(f', g') \mathcal{F}(M)^*(f'', g;;) \mathcal{T}(f', g'; f'', g'')$$

$$\exp(-j2\pi[(f' - f'')x' + (g' - g'')y']) df' dg' df'' dg'', \quad (21)$$

where $H$ denotes the optical transfer function (OTF) of the projector system; $\mathcal{F}(M)$ represents the mask pattern $M$ transformed by fast Fourier transform (FFT); $M^*$ is the complex conjugate of the mask; $(x', y')$ denotes the spatial coordinates on the image plane; $(f, g), (f', g')$ and $(f'', g'')$ represent the normalized frequency-domain coordinates.

The most important term is $\mathcal{T}$ the *transmission cross coefficient* (TCC), $\mathcal{T}$, which encodes how the source distribution $J$ couples with the pupil transfer function $H$ to form image intensity.

$$\mathcal{T}(f', g'; f'', g'') = \iint J(f, g) H(f + f', g + g') H^*(f + f'', g + g'') df dg. \quad (22)$$

To approximate the TCC efficiently, we use the SVD-based *sum of coherent sources* (SOCS) decomposition:

$$\mathcal{T}(f', g'; f'', g'') = \sum_{i=1}^{r} \alpha_i h_i(f', g') h_i^*(f'', g''), \quad (23)$$

where $\alpha_i$ and $h_i$ are the $i$-th eigenvalue and eigenfunction of the TCC.

With convolution theroem, the Hopkins intensity reduces to a sum of coherent images:

$$I(x, y) = \sum_{i=1} \alpha_i \left| (h_i * M)(x, y) \right|^2$$

$$= \sum_{i=1} \alpha_i \left| \mathcal{F}^{-1}\big( \mathcal{F}(h_i) \odot \mathcal{F}(M) \big)(x, y) \right|^2, \quad (24)$$

where $*$ denotes spatial convolution and $\odot$ is pointwise multiplication.

This description of the Hopkins model follows (Yang et al., 2022a; Chen et al., 2024).

# E  EXPERIMENTAL DETAILS

## E.1  DETAILS OF EVALUATION METRICS IN SEMICONDUCTOR COMPUTATIONAL LITHOGRAPHY

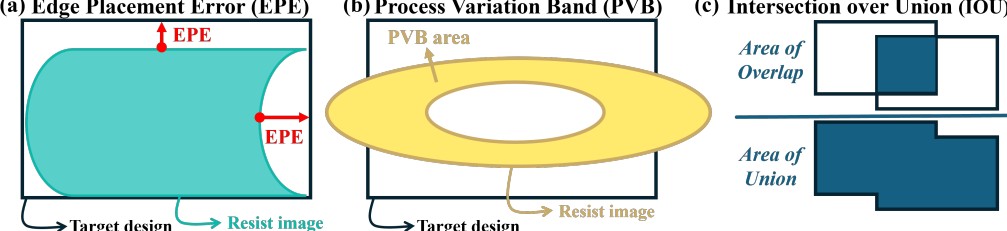

Figure 11: Illustrations of various evaluation metrics in computational lithography: (a) Edge Placement Error (EPE), which quantifies the deviation of edges from their expected positions; (b) Process Variation Band (PVB), showing the tolerance range of variations; and (c) Intersection over Union (IOU), measuring the overlap accuracy between predicted and reference regions.

In this subsection, we provide detailed explanations of the evaluation metrics employed in our study—Edge Placement Error (EPE), Process Variation Band (PVB), and Intersection over Union (IOU)—clarifying their definitions, significance, and roles within the context of computational lithography. As shown in Figure 11 (a), EPE quantifies the deviation between the intended edge positions of the target design on the mask and the actual edge positions observed in the resist image resulting from the computational lithography process. In other words, this metric is expressed as the distance between the predicted and target edge positions. In chip manufacturing, even minor deviations in edge placement can result in significant functional or performance degradation in the fabricated semiconductor devices. As a result, EPE serves as a vital indicator of whether a given mask design can be reliably reproduced with the required level of precision. On the other hand, PVB represents the range of possible edge positions that may arise due to variations in the manufacturing process. These variations stem from factors such as exposure dose, focus shifts, and resist chemistry fluctuations. As depicted in Figure 11 (b), PVB is typically visualized as a band encompassing the range of edge deviations under different process conditions, thereby offering insights into the robustness of a given mask design. A smaller PVB indicates a mask design that is less sensitive to process variations, which is essential for ensuring yield and performance consistency under diverse manufacturing conditions. Finally, IOU is a widely used evaluation metric in image segmentation tasks. In computational lithography, it is employed to assess the spatial overlap between predicted resist images and their corresponding true reference regions. Specifically, as illustrated in Figure 11 (c), IOU is defined as the ratio of the intersection area to the union area of the predicted and ground truth resist regions. This metric provides a holistic measure of the accuracy with which the predicted patterns replicate the intended design.

### E.2 Hyperparameters and Training Strategies

#### E.2.1 Overall Training Procedure of Mask Optimization

Our method follows the standard training methodology provided by LithoBench (Zheng et al., 2023b) for both mask optimization and lithography simulation tasks. This approach has been uniformly applied to all models presented in our experiments to ensure fair comparison. For the mask optimization task, LithoBench includes a **pretraining stage**, since untrained mask optimization models tend to generate blank simulated resist images. We follow this procedure as it also significantly reduces computational overhead during training.

Briefly, the difference between pretraining and training lies in the labels used for the MSE calculation.

**Pretraining stage.** Given a target wafer pattern (image) $R^*$, LithoBench provides a reference mask $M_{\text{ref}}$. During the pretraining stage, the mask optimization model is trained to generate a mask $M$ that follows this reference mask $M_{\text{ref}}$ using the MSE loss:

$$\mathcal{L}_{\text{pretrain}} = \mathcal{L}_{\text{mse}}(M, M_{\text{ref}}). \tag{25}$$

**Training stage.** The generated mask $M$ passes through a lithography simulator $g(\cdot)$ to obtain a simulated resist image $g(M)$. The model is trained to minimize the difference between this simulated resist image and the ground-truth target wafer pattern (image) $R^*$.

$$\mathcal{L}_{\text{training}} = \mathcal{L}_{\text{mse}}(g(M), R^*). \tag{26}$$

#### E.2.2 Hyperparameters

To ensure a fair comparison, all competing methods adopt the same training procedures and hyperparameters. For MetalSet, we employ 2 epochs of pretraining, then 8 epochs of training, with a batch size of 8. For ViaSet, we adopt 1 epoch of pretraining, then 2 epochs of training, again using a batch size of 8. In the OP kernel, we set the kernel size to 11. We consistently use the default learning rate, its scheduling, and additional hyperparameters from LithoBench to maintain fair comparisons across different models.

#### E.2.3 Implementation Details of ILILT

We incorporate a CFNO backbone to ILILT but refine the iterative unrolling procedure to reduce computational overhead. Rather than summing unrolled losses starting at step $T/2$, we update its model at every unrolling step, lowering cost and accelerating training. Also, we do not apply an

Table 11: Hyperparameter configurations.

| Setting | Hyperparameter | Configuration |
|---------|----------------|---------------|
| MetalSet | Pretrain epochs | 2 |
| | Train epochs | 8 |
| ViaSet | Pretrain epochs | 1 |
| | Train epochs | 2 |
| Default settings | Batch size | 8 |
| | Optimizer | Adam |
| | Learning rate | 0.001 |
| | Learning rate decay | 0.1 |
| | Learning rate decay policy | Step |
| | Learning rate decay epoch | Train epochs // 2 |

exponential weighting to the loss across different unrolling depths, we update its loss directly at each unrolled step. In our experiments, we use the unrolling depth to $T = 2$.

### E.2.4 COMPUTATIONAL COST

We present additional analysis on computational cost. Our approach achieves significantly faster speeds than the current state-of-the-art ILILT model. Moreover, adding the OptiCo block to our backbone does not noticeably increase the overall computational cost. In fact, OptiCo requires less overhead compared to incorporating an FNO-based module, demonstrating a favorable trade-off between accuracy and efficiency. All experiments were conducted using PyTorch (Paszke, 2019) with an NVIDIA RTX 3090 GPU, except for ILILT, which was tested on an NVIDIA A6000.

Table 12: Training and inference time comparison across models.

| Model | Training (s / epoch) | Inference (s / img) |
|-------|----------------------|---------------------|
| DOINN | 1793 | 0.107 |
| CFNO | 2246 | 0.129 |
| DAMO | 5930 | 0.149 |
| ILILT | 21237 | 0.441 |
| Backbone | 6954 | 0.138 |
| Backbone+FNO | 8412 | 0.162 |
| Backbone+OptiCo (Ours) | 7880 | 0.151 |

## F RELATED WORK

### F.1 MASK OPTIMIZATION

Mask optimization in computational lithography is the process of iteratively refining the mask layout to ensure that the patterns printed on the wafer closely match the intended design. This process can be viewed as the inverse of lithography simulation: rather than predicting the resist image (wafer pattern) from a given mask, mask optimization starts with the desired wafer pattern and works backward to compute the mask design that will produce that pattern under the constraints of the lithography process. Similar to LithoGAN's (Ye et al., 2019) pioneering use of GAN for lithography simulation, GAN-OPC (Yang et al., 2018) emerged as one of the early DL methods that employ a conditional GAN architecture to improve mask optimization. Following this, Neural-ILT (Jiang et al., 2020) proposed a fine-grained, end-to-end correction framework based on the UNet architecture, which significantly improved the mask optimization process. Subsequently, L2O-ILT (Zhu et al., 2023) proposed an innovative approach by embedding iterative optimization algorithms into the neural network architecture, enabling the generation of high-quality masks. More recently, reinforcement learning (RL) algorithms have been actively applied to mask optimization frameworks to achieve superior performance. RL is particularly well-suited for this mask optimization task because it can

optimize non-differentiable objectives by utilizing feedback in the form of rewards, allowing the model to learn optimal policies through trial and error (Shin et al., 2024a). For instance, RL-OPC (Liang et al., 2023) and CAMO (Liang et al., 2024) have successfully incorporated RL algorithms into their frameworks and demonstrated outstanding performance by leveraging the exploratory capabilities of RL agents to navigate the complex design space.

## F.2 DIFFRACTION-INSPIRED MASK OPTIMIZATION TECHNIQUES

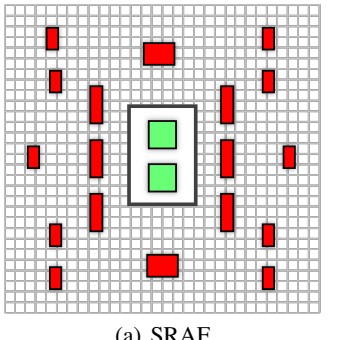 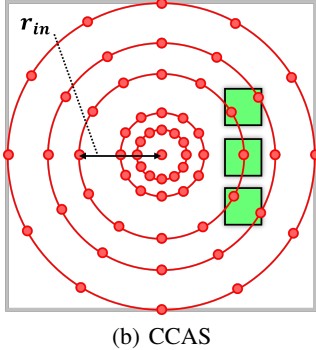

(a) SRAF                    (b) CCAS

Figure 12: Visualization of diffraction-inspired mask optimization techniques. (a) Sub-Resolution Assist Features (SRAF) and (b) Concentric Circle Area Sampling (CCAS). Green regions indicate the target wafer design patterns.

As technology nodes continue to shrink, diffraction effects become increasingly significant in lithography, motivating the widespread use of Sub-Resolution Assist Features (SRAFs) to capture and compensate for these effects (Xu et al., 2016). SRAF refers to assist blocks placed around the main pattern to mitigate diffraction-induced errors and improve pattern fidelity. In parallel, the Concentric Circle Area Sampling (CCAS) layout (Matsunawa et al., 2016) also draws inspiration from diffraction, injecting prior knowledge into the mask optimization process via a circular, pre-defined feature extractor. This diffraction-inspired feature extractor is designed to improve SRAF placement by explicitly accounting for wave-based propagation characteristics (Yang et al., 2022b). Notably, SRAF corresponds to the diffraction-inspired rings discussed in the main text.

## F.3 LITHOGRAPHY SIMULATION

Lithography simulation is the computational process of modeling and predicting the performance of a lithographic system, assessing how effectively a mask pattern transfers onto silicon wafers. In recent years, a variety of machine learning (ML) and deep learning (DL) methods have been developed to advance computational lithography simulation (Zheng et al., 2023b). One of the pioneering frameworks in this domain is LithoGAN (Ye et al., 2019), which was among the first DL models to utilize a Generative Adversarial Network (GAN) for creating a direct correspondence between input mask designs and their resulting wafer patterns. Building on the foundation laid by LithoGAN, DAMO (Chen et al., 2020) introduced enhancements by incorporating a UNet++ backbone (Zhou et al., 2018; Ronneberger et al., 2015; Wang et al., 2018) and residual blocks into its architecture. These modifications allowed DAMO to achieve higher-resolution mask predictions, thereby improving the fidelity and precision of the simulated wafer patterns. Further advancing the field, TEMPO (Ye et al., 2020) addressed the challenges associated with 3D masks in Extreme Ultraviolet (EUV) lithography. To be more specific, TEMPO employed multi-domain image-to-image translation techniques to accurately predict the intensity of aerial images at varying resist heights. In addition to these developments, recent studies have explored the application of Fourier Neural Operators (FNO) (Li et al., 2020) to enhance lithography simulation. DOINN (Yang et al., 2022a) was one of the first to introduce an FNO-based framework, utilizing Fourier transforms to capture both low-frequency global information and high-frequency local details. Building upon this success, CFNO (Yang & Ren, 2023) further refined and advanced the approach by integrating inductive lithography biases into the model architecture inspired by ViT (Dosovitskiy, 2020).

# G  BACKBONE NEURAL NETWORKS

In this section, we provide an overview of backbone neural networks used in computational lithography, discussing their general role and emerging trends in the field. Additionally, we present detailed descriptions and visualizations of the specific backbone architectures employed in our study.

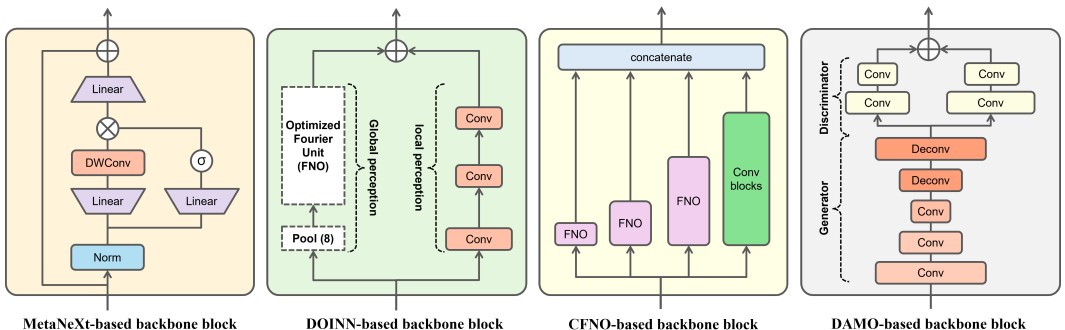

Figure 13: Visualizations of the backbone neural network used in our study, including MetaNeXT, DOINN, CFNO, and DAMO.

Backbone neural networks play a critical role in both lithography simulation and mask optimization by extracting features and encoding information from input mask or resist images. Traditionally, convolutional neural networks (CNNs) have been the backbone of choice in these tasks due to their inherent ability to capture spatial hierarchies and intricate patterns within images. In particular, CNNs leverage localized receptive fields and weight sharing, which not only enhance their computational efficiency but also enable them to recognize and process spatial features effectively. However, despite their strengths, CNNs face limitations in capturing global context and long-range dependencies within data. The localized nature of convolutional filters restricts their ability to integrate information across distant regions of an image. To address these challenges, Fourier Neural Operators (FNOs) have been introduced as an alternative backbone architecture. FNOs operate in the Fourier domain, where they can efficiently model global patterns and capture long-range dependencies by transforming spatial data into the frequency domain (Li et al., 2020). In addition to FNOs, transformer (Vaswani, 2017) architectures have emerged as a powerful option for backbone networks. Specifically, transformer architectures utilize self-attention mechanisms, which enable them to weigh the significance of different parts of the input data dynamically. This capability allows the transformer to capture global dependencies and intricate relationships within the data (Son et al., 2024). Recently, Mamba architectures (Gu & Dao, 2023), which integrate spatial attention with efficient computation, have emerged as a compelling alternative to transformers, and they also show promise as potential backbone networks.

As shown in Figure 13, we present visualizations of the backbone neural networks used in our study, including MetaNeXt, DOINN, CFNO, and DAMO. Each architecture employs distinct mechanisms for feature extraction and representation. Notably, MetaNeXt utilizes depth-wise convolution (DW-Conv) blocks (Liu et al., 2022; Yu & Wang, 2024), which efficiently capture spatial features by applying independent convolutional filters to each channel. This lightweight yet powerful design enables MetaNeXt to extract fine-grained details while maintaining computational efficiency. Due to its simplicity and effectiveness, MetaNeXt serves as our primary backbone neural network. On the other hand, DOINN and CFNO integrate Fourier Neural Operator (FNO) blocks, which operate in the Fourier domain to implicitly model light behavior. By transforming spatial data into frequency space, these architectures effectively capture both global and local features, improving their ability to model lithographic processes. Given the promising aspects of FNO-based approaches in computational lithography, we conducted additional experiments to evaluate the versatility and compatibility of OptiCo when applied to these architectures. Our results confirm that OptiCo is not limited to a single backbone but can be seamlessly incorporated into various network architectures, including FNO-based backbones like DOINN and CFNO, further demonstrating its adaptability and effectiveness in computational lithography.

# H  LIMITATIONS AND FUTURE STUDY

In this paper, we focused on lithography modeling by incorporating fundamental physical principles through the Fresnel diffraction and demonstrated strong performance in both lithography simulation and mask optimization tasks. However, we think several avenues exist for further improving the mask optimization task. Future work could investigate advanced optimization techniques, such as progressive optimization (Shin et al., 2024b), inverse optimization (Byrne, 2014; Yang & Ren, 2025), Pareto optimization (Qian et al., 2015), reducing mini-pixel errors (Yang et al., 2024), and data augmentation (Liu et al., 2023) to address the inherent complexities of the mask optimization task in computational lithography. Another promising direction is to consider source-level modeling jointly with mask optimization (Ma et al., 2025; Harrison, 2025), which could further improve fidelity under real optical conditions and interferograms (Jeong et al., 2025).

