# OpenReview forum: "Optical Diffraction-based Convolution for Semiconductor Mask Optimization"
_ICLR.cc/2026/Conference — ICLR 2026 Conference Withdrawn Submission_

### Official Review · Reviewer_Gw2z · 2025-10-29

**Soundness:** 3
**Presentation:** 3
**Contribution:** 3
**Rating:** 6
**Confidence:** 5

**Summary:**

This paper proposed a genAI-based framework for ILT tasks. The major contribution is the embedded diffraction kernel that reflects light propagation. Experiments on lithobench show superior performance over existing genAI-based solutions. The proposed kernel can also serve as a plug-in to boost the performance of existing models.

**Strengths:**

This paper comes with a good presentation and reasonably sound methodology supported by complete experiments. In detail:
1. Introduce the optical diffraction kernel to introduce physics bias in the model pipeline.
2. Good performance over SOTA AI-based solutions with a large margin.
3. Complete ablation study shows the effectiveness of each technique's contributions in the architecture development.
4. The overall paper is clearly written with only a few things that need to be addressed.

**Weaknesses:**

1. AI-based ILT has long been studied; however, in practical foundries never trust a pure-AI-based approach due to non-deterministic results and its statistical nature. I wonder how the proposed model behaves compared to numerical solvers. There are many ILT solvers come out recently. e.g. Multi-ILT [Sun+, DAC'23], CurvyILT [Yang+, ISPD'25].
2. I would assume the EPE numbers are based on a 15nm threshold rule. However, this number is way too high for practical usage. I wonder if it is possible to show the EPE values.
3. How does the model scale to a larger layout?
4. The optical diffraction kernel is based on the forward lithography imaging. Introducing the kernel benefits the inverse application is somehow against the common intuition. It's better to provide some explanation.

**Questions:**

refer to the weakness.

---

### Official Review · Reviewer_ZUad · 2025-10-29

**Soundness:** 1
**Presentation:** 2
**Contribution:** 1
**Rating:** 2
**Confidence:** 4

**Summary:**

The authors propose a new neural network structure that integrates optical diffraction for mask optimization in semiconductor design. The core layer of the neural network is a complex-valued convolution modulated by an analytic diffraction kernel, which combines with a MetaNeXt backbone and eventually maps a target resist pattern to a mask design. The proposed network outperforms other methods and generalizes better to unseen data.

**Strengths:**

The authors did a reasonable comparison against existing methods and presented a bunch of ablation studies.

**Weaknesses:**

The neural network structure design doesn’t really follow the physics and optical diffraction, which makes the intuition confusing and the contribution claim misleading.

**Questions:**

1. How is the lithography simulation being performed? The input and output in this case are exactly opposite of the mask optimization; are they using the same model?
2. While I appreciate the physics intuition of using the propagation kernel in the convolution, the way that it is used in the model is confusing. The propagation here is from the mask plane to the wafer plane, but the kernel here maps from a target wafer image to a mask design. There should be more explanation on how the model design corresponds with the optical propagation, rather than listing the well-known derivation of convolution in optical diffraction.
3. It is also not clear why the output is the addition of a ‘physics module’ and a backbone model. There seems to be no underlying physics to this operation.
4. If the author makes the OP kernel learnable with rotational symmetric constraint, will it converge to the original OP? And what are the differences between the four OP kernels shown in Figure 4?
5. Why would a more accurate physical expression (e.g., a larger kernel size, Helmholtz-based kernel) lead to a worse result if the model is explicitly modeling the physical process? This echos with my second question that the underlying physics is not clear.

---

### Official Review · Reviewer_coaE · 2025-10-30

**Soundness:** 2
**Presentation:** 3
**Contribution:** 2
**Rating:** 4
**Confidence:** 4

**Summary:**

This paper proposes OptiCo, a novel CNN architecture for computational lithography that aims to incorporate physical principles of optical diffraction directly into the network. The authors identify a relevant gap in current deep learning approaches for lithography, namely the lack of explicit physical modeling. The paper is well-written, and the experimental results appear strong, demonstrating promising performance and generalization.

**Strengths:**

(1) Well-motivated problem statement highlighting a genuine shortcoming in existing literature.

(2) The proposed OptiCo model is novel in its attempt to hard-code physical knowledge.

(3) Comprehensive experiments showing superior performance on benchmarks and OOD datasets.

(4) Clear and professional presentation.

**Weaknesses:**

The most significant concern with this work is a potential fundamental confusion regarding the forward and inverse problems in computational lithography.

The paper correctly states that optical diffraction is the physical process that transforms a mask pattern into an aerial image (the forward model). However, the task of mask optimization is inherently an inverse problem: finding a mask that produces a desired aerial image or wafer pattern.

The authors propose using a convolutional kernel to model optical diffraction (a forward process) within a network designed for the inverse problem. This creates a conceptual mismatch. For inverse design, the network must learn to invert or compensate for the physics of diffraction, not merely simulate it. Embedding a forward model kernel directly into an inverse problem solver without a clear architectural or loss-based mechanism to handle the inversion is conceptually misleading and may limit the model's effectiveness. The authors need to clarify how the inclusion of a forward physical model directly aids in solving the inverse problem, as this is the core of their contribution.

**Questions:**

See Weaknesses

---

### Official Review · Reviewer_E2pw · 2025-11-01

**Soundness:** 2
**Presentation:** 3
**Contribution:** 1
**Rating:** 2
**Confidence:** 4

**Summary:**

This paper proposes OptiCo, another new physics-guided CNN that embeds an optical phase (OP) kernel with complex convolutions for lithography mask optimization and simulation, showing lower MSE/EPE and some OOD gains on LithoBench with a TV prior for manufacturability.

**Strengths:**

- Computational lithography is impactful for future computation and accelerating accurate simulation and optimization matters.

- Encodes diffraction via an OP kernel shows empirical gains with reported improvements on benchmark tables.

**Weaknesses:**

Overall, I do not find this paper is interesting as the architectural change is domain-specific and incremental in the long line of physics-informed models and there isn’t a broadly useful for ML community. For a top ML venue, the ML contribution is weak the work seems better aligned with EDA venues where domain impact is the primary metric. If considering its contribution to AI for science and engineering, the real-world experiments would be required to justify its impact.
Moreover, the evaluation cannot rule out cherry-picking with unclear splits/model selection, single-seed reporting, limited robustness and results may be benchmark and simulator-locked.

- Limited ML novelty & generality. OP-kernel + complex convs is an incremental tweak as in previous lines of computational lithography papers they already have considered similar ideas. It is also task-specific and lacks a general learning insight likely to interest the broader ML community.
- The experiment has unclear train/val/test protocol with no precise split definitions,  ambiguous use of validation for early stopping/hyper-params and appropriate usage of test-set is not guaranteed.
- They almost use single-seed results; there are no mean±std or significance tests which cannot rule out cherry-picking risk.
- OOD protocol is under-specified. Construction and difficulty balance of OOD sets are unclear. The stability across different split draws not shown.
- The method is quite empirical. There is no theoretical insight on why the proposed architectural changes can be effective for lithographic modeling. This problem is non-negligible especially when demonstrating only on benchmarks.
- The method is tied with specific simulator and process. No sensitivity to source/resist/NA changes or alternate simulators; gains could reflect over-tuning to one stack.
- The ablations need to be cleaner. Right now it’s hard to tell what’s driving the gains. Please keep the training budget and seeds the same across variants, and pick checkpoints by validation, not by best test score. Also separate the effects such as show what changes when you switch to complex convs, when you change kernel size, and when you add TV.
- The authors need to make baseline comparisons fairer. Tell us the exact hyperparams and training time you used for each baseline, and try to match hardware and budgets. If one model gets more epochs or a faster GPU, efficiency and accuracy claims won’t be comparable.

**Questions:**

Same as weakness.

---

### Note · Authors · 2025-11-12

**Comment:**

Thank you for your reviews.

**Withdrawal Confirmation:**

I have read and agree with the venue's withdrawal policy on behalf of myself and my co-authors.